# Is the Impact of Food Insecurity on Food-Related Behavior Moderated by Generation in South Korea?

**DOI:** 10.3390/ijerph22050766

**Published:** 2025-05-13

**Authors:** Hyo Sun Jung, Yu Hyun Hwang, Hye Hyun Yoon

**Affiliations:** 1Center for Converging Humanities, Kyung Hee University, Seoul 02447, Republic of Korea; chefcook@khu.ac.kr; 2College of Hotel & Tourism Management, Kyung Hee University, Seoul 02447, Republic of Korea; yhhwang@khu.ac.kr

**Keywords:** perceived food insecurity, food purchasing behavior, food preparation practices, life satisfaction, generations

## Abstract

The purposes of this study were to determine whether perceived food insecurity significantly influences food purchasing behaviors, food preparation practices, and life satisfaction; and to verify the moderating effect of generation in South Korea. Structural equation modeling and multi-group analyses were conducted to confirm the research model and test the hypotheses. Perceived food insecurity positively influenced food purchasing behaviors and food preparation practices. In addition, this study verified the presence of a significant negative relationship between perceived food insecurity and life satisfaction. Notably, a greater awareness of food insecurity in Generation M increased food preparation practices, and the negative impact of perceived food insecurity on life satisfaction was more strongly exhibited in the older generation than in the younger generation. This presents a need to pay greater attention to food insecurity issues at the government level and accelerate related future research. Specifically, it will be necessary to establish a structured educational environment and support services for systematic food security education. Also, rather than educating people with high levels of perceived food insecurity about why they should eat healthy food, proactive efforts should first be made to remove barriers to healthy eating habits. As a preventive measure against the worsening of food insecurity, more solutions should be prepared from a macroscopic perspective in addition to implications from the microscopic perspective mentioned so far.

## 1. Introduction

In times past, food insecurity, which had a significant influence on human life, was a social and economic issue exhibited only in vulnerable groups [1,2]. However, many social and environmental events threatening secure food systems continue to occur globally today [3,4]. In particular, with restricted food exchange and trade between countries due to quarantine and border closures under the COVID-19 crisis, food prices have increased, while food accessibility has gradually lowered [5]. The COVID-19 outbreak has brought hunger to millions of people around the world [6]. In many countries, the COVID-19 pandemic has limited the activities of food industry workers as food factory production has decreased [7]. In fact, studies from the early months of the pandemic found that food insecurity had more than tripled compared to the previous year [8]. Food insecurity is exacerbated by stressors and shocks, such as food crises, rising inequality, economic instability, or pandemics like COVID-19 [9]. Even after COVID-19, food insecurity remains a global challenge, still affecting a significant portion of the world’s population [10].

Previous studies have proven that people experiencing food insecurity generally exhibit unhealthy eating habits [11]. This indicates that food insecurity can also occur when there is inadequate physical and economic access to the necessary food [12]. Therefore, it is evident that food insecurity immensely affects not only personal mental and physical well-being but also affects a person’s overall quality of life [13]. Eventually, food insecurity can manifest as unhealthy eating habits and a serious public health problem, which can lead to poor eating behaviors and chronic diseases [14]. Cox et al. [15] suggest that awareness of food insecurity itself elevates the risk of developing serious chronic diseases. On the other hand, people who have secured food without awareness of food security generally have strong confidence and autonomous behavior and are highly satisfied with their lives [16]. In particular, food insecurity tends to occur simultaneously with other socioeconomic problems such as housing instability and job instability, and it is said to have a very meaningful connection with the current social phenomenon in South Korea [17]. In recent years, depression and mental illness have been increasing in South Korea, and income poverty has also continued, so concerns about food insecurity are increasing [18].

Nevertheless, there is currently very little understanding of the food crisis and its impact on food insecurity, as well as food purchasing behaviors and food preparation practices. Therefore, solid evidence is necessary to encourage policy measures or development to assist in selecting healthier foods. However, existing research is considered inadequate for observing the elaborate relationships among the above research constructs. Therefore, comprehensive research is required to clarify the impact of these variables on food insecurity, and the resulting data will be useful in helping to find practical ways to actively respond to the food crisis. In this context, this study aimed to determine whether perceived food insecurity significantly influences food purchasing behaviors, food preparation practices, and life satisfaction and whether this relationship is moderated by the variable of generation (Figure 1).

## 2. Literature Review and Hypothesis Development

### 2.1. Perceived Food Insecurity

Food insecurity, which refers to the “limited availability of nutritionally adequate food or uncertain abilities to obtain safe food in a socially acceptable manner” [19], causes serious short- and long-term psychological and physical problems in people’s lives [20]. Notably, because households perceiving food insecurity in health and physical aspects face chronic food shortages [21,22], they are generally in a state of nutritional imbalance that causes health problems [23]. In addition, many people without food shortages overly rely on processed or energy-dense foods due to their limited food budgets [24,25]. These unhealthy eating habits lead to various adult diseases in individuals (e.g., obesity, diabetes, and even cardiovascular diseases), thereby destroying their physical condition [26,27]. Moreover, food insecurity not only has a negative impact on physical health but also results in serious problems in psychological and mental wellness [28]. People suffering from food insecurity also experience anxiety, tremendous stress, fear, and even suicidal thoughts under financial pressure [29]. In addition, when they fail to intake sufficient nutrition and fulfill their appetites, they tend to suffer from mental diseases, including severe depression, which is often exhibited as self-destructive or abnormal behavior [30,31]. Therefore, it is abundantly clear that food insecurity reduces individuals’ overall quality of life as well as the physical and mental well-being of individuals and creates a negative linkage effect [32,33]. Eventually, issues associated with food insecurity concern the stability of families, communities, and society as a whole [34]. Therefore, many researchers from various fields around the world are paying great attention to the important assignment of food security and the severity of the food insecurity issue [35]. Governments have also made efforts to find efficient solutions to mitigate this serious problem [36].

### 2.2. Relationship Between Food Insecurity and Food Purchasing Behaviors

Stone et al. [37] stated that food insecurity leads to certain purchasing behaviors related to food and particularly restricts people’s budgeting and supermarket use. They confirmed that food insecurity changes people’s dietary life-related purchasing behaviors. Adams [38] also reported that people experiencing food insecurity use discounts and coupons to increase their purchasing power. In addition, Kim et al. [39] and Rasmusson et al. [40] noted that food insecurity causes people to develop the behavior and habit of purchasing affordable and inexpensive foods that are harmful to their health. Chang et al. [41] argued that people who experience food insecurity buy products without regularly checking nutrition labels. Begley et al. [42] also reported that people who become aware of food insecurity are less likely to use food labels, plan ahead for meals, and make a shopping list. Also, Knol et al. [43] highlighted that people who experience food insecurity are more likely to purchase processed foods than those who do not. Based on these research findings, the following hypothesis was established:

**Hypothesis** **1.**
*Perceived food insecurity positively influences food purchasing behavior.*


### 2.3. Relationship Between Food Insecurity and Food Preparation Practices

Gorton et al. [44] showed that people who experienced food insecurity had lower levels of culinary skills than people who were food-secure. Malan et al. [45] found in a study on college students suffering from food insecurity that students perceived the need to enhance their food cookery skills. Soldavini et al. [46] noted that people experiencing food insecurity have lower confidence in their cooking skills than those without food insecurity. Pepetone et al. [47] reported that food insecurity leads to limited health-related knowledge and passive attitudes in the process of food preparation. Armstrong et al. [48] demonstrated that people who did not experience food insecurity used a wider range of cooking techniques, as well as prepared and used food with greater confidence than those who experienced food insecurity. According to Miller et al. [49], when people experience food insecurity, their meal preparation skills become highly limited due to excessive demands on their time, limited financial resources, and a lack of motivation to follow a healthy diet. Moreover, people with food insecurity are prone to select batch cooking methods [50], use energy-saving home appliances [51], and prepare foods in unhealthy forms of packaging, with a high starch content [52]. According to Hevesi et al. [53], parents struggling with food insecurity protect their children from hunger by eating less themselves, relying on free school meals, and seeking help from their families and food banks in times of food crisis. This suggests that food insecurity can also influence food preparation in raising children. Based on these research findings, the following hypothesis was established:

**Hypothesis** **2.**
*Perceived food insecurity positively influences food preparation practices.*


### 2.4. Relationship Between Food Insecurity and Life Satisfaction

Life satisfaction is the degree to which an individual is satisfied with different areas of their life [54], and life satisfaction is another important factor intertwined with food insecurity [55]. Food insecurity appears to be linked to lower levels of life satisfaction, happiness, quality of life, and overall well-being [56,57,58]. Grunert et al. [59] observed the serious side effects of food insecurity on personal life. According to researchers such as Holm et al. [60] and Salahodjaev and Mirziyoyeva [61], food insecurity caused by high food prices and limited access to a stable food environment becomes a large barrier to fulfilling healthy eating habits and, as a result, has a heavily negative effect on people’s life satisfaction. Ahmadi et al. [62] stated that food insecurity is directly related to satisfaction in life, and Selvamani et al. [63] defined food insecurity as an important factor in determining quality of life and satisfaction in elderly people. According to Jung et al. [55], there are important factors that affect satisfaction in life: food security, subjective health conditions, and living environment satisfaction. Among them, food security is the most significant factor. Based on these research findings, the following hypothesis was established:

**Hypothesis** **3.**
*Perceived food insecurity negatively influences satisfaction in life.*


### 2.5. Moderating Role of Generations

A generation of people develop similar tendencies [64] as they go to school, get a job, retire, and experience memorable historical events in this process [65]. The Baby Boomer generation was born between 1950 and 1964 and is accustomed to collectivism, whereas Generation X was born along with the birth of individualism. Generation M, which is the so-called “Millennial Generation”, born between 1980 and 1996, is also the first generation to emerge in the Internet era and can be described as a generation with a high propensity for self-expression. Ahn et al. [66] assessed that food insecurity can increase in older people, which is not irrelevant to their reduced economic power with age, and Jackson et al. [67] reported that food insecurity among elderly people is high and can disrupt their ability to obtain, prepare, or consume adequate amounts of food [68]. According to Leung and Wolfson [69], in addition to the fact that food insecurity among the elderly continues to increase, both the unhealthy dietary index and dietary score appear to be low in elderly people. There is also a study showing that food insecurity among middle-aged people can be more intense than before [70]. Moreover, according to a recent consumer food survey by Lusk and Polzin [71], the proportion of adult members of Generation Z who experience food insecurity was more than double that of average Americans, and in fact, one in three Americans born between 1996 and 2004 experienced difficulty procuring sufficient food in 2022. This proportion is worth comparing with less than 1 in 5 members of the Millennial Generation and Generation X and less than 1 in 10 members of the Baby Boomer generation, as shown in the following survey. A study by Sahin and Celik [72] demonstrated that single young women had a greater awareness of food insecurity than married women. As this shows, there are various inconsistent results in the available literature regarding the effects of age on perceived food insecurity. However, given that there are clear differences in the perception of food insecurity among generations, the following hypothesis was established based on the research findings thus far:

**Hypothesis** **4.**
*The influence of perceived food insecurity on food purchasing behavior, food preparation practices, and life satisfaction will vary across generations.*


## 3. Methodology

### 3.1. Procedures

In this study, a survey was distributed to general consumers, and the survey was collected through an online data collection company called Embrain (Embrain Company, Seoul, Republic of Korea). A stratified sampling method was used to ensure that various age groups were included evenly. The English questionnaire was translated from the original questionnaire into Korean, and then translated back into English to ensure that there were no notional differences between the two versions [73]. Prior to this survey, a pilot survey was conducted on 50 samples. Based on the results of this pilot test, we used revised questions that we thought were difficult or ambiguous. In total, 500 samples were allocated over a two-week period from 1 to 14 May 2024. The participants were guaranteed that their collected data would be kept secure. A total of 300 questionnaires were utilized for the final analysis.

### 3.2. Measurement of Variables

To ensure the use of measurement variables with high reliability and validity, this study adopted variables that had been sufficiently verified in existing studies. The variables used in this study were specifically grouped into five categories (food insecurity, food purchasing behavior, food preparation practices, life satisfaction, and demographic characteristics), and the four variables excluding demographic characteristics were measured on a seven-point Likert scale. To measure consumer awareness of food insecurity, 7 questions developed by Myers [74] were employed, and the participants were asked to answer how often they experienced these items (e.g., Worry about food, and Unable to eat preferred foods). Food purchasing behavior was measured by dividing it into budgets and supermarket offers and using a total of 10 questions (e.g., Bought smaller amounts of dried goods (pasta, lentils) so I only buy what I need, and Bought more own-brand food and drink) with reference to a study by Stone et al. [37]. In addition, food preparation practices were divided into meal planning and resourcefulness to measure, with a total of 6 questions (e.g., Plan all meals for the week in advance, and Reduced the amount of food that I wasted) [30]. Life satisfaction was measured with a total of 3 questions (e.g., I am satisfied with my life) by referring to a study by Suldo and Huebner [75]. Lastly, demographic information on the respondents was surveyed using three questions regarding gender, age, education level, and marital status.

### 3.3. Analysis Methods

For the analysis, we implemented structural equation modeling (SEM), which consists of a two-step procedure of measurement model and structural model analysis [76]. Measurement model analysis was conducted to examine the reliability and discriminant validity of the study variables to find the basis for the structural relationships in the model. In the following structural model analysis, we evaluated the hypothesized model and estimated parameters. Moreover, a multi-group analysis was carried out to understand any differential influences among generations [77]. A Harmon test was also performed to verify any CMB (Common Methods Bias) errors that could occur when the samples were collected in parallel with the data collection process.

## 4. Results

Table 1 presents the profiles of the samples. By gender, 48.0% were female, and 52.0% of the subjects were male. By generation, Generations M and X accounted for 38.7% and 35.3% of the subjects, respectively. In terms of educational levels, those with a college degree or above represented the largest portion, at 66.0%. Regarding marital status, married people accounted for 55% of the total.

The CFA included a total of 26 items, which were then factorized into six items, and the fit of the model was sufficient (χ^2^ = 633.999; df = 284; χ^2^/df = 2.232; RMSEA = 0.064; IFI = 0.949; GFI = 0.858; TLI = 0.941; CFI = 0.948).

Table 2 contains the analysis results for the measurement model. The standardized estimates of all items were at least 0.6, and the t-values were also higher than 10.0 (*p* < 0.001) [78]. In addition, the composite reliability estimate for each variable exceeded 0.7 (0.708~0.925), and the AVE values also exceeded 0.5 (0.597~0.828), confirming that internal consistency was met. The Cronbach’s alpha value derived from the reliability analysis also exceeded 0.8 [79]. Table 3 presents the results of a correlation analysis between constructs, which confirmed that the relationships between all constructs were in the same direction as the hypothesis. Given that the study data were collected using a self-report questionnaire, Harmon’s single-factor test was conducted to identify CMB errors. According to the test results, the variance of a single factor (39.501%) did not account for more than half of the total cumulative variance (79.159%), and there was no serious bias because no factor that accounted for most of the covariance was included in the measurement items.

In the next step, the relationships between the variables of the proposed model were verified using a structural equation model (SEM), and the goodness of fit of the model (χ^2^ = 715.164; df = 294; χ^2^/df = 2.433; GFI = 0.838; IFI = 0.938, CFI = 0.938; RMSEA = 0.069) was confirmed to be appropriate. Table 4 presents the estimated parameters, t-Value, and *p*-value for the relationships between the proposed model. Hypothesis 1 established that perceived food insecurity positively influences food purchasing behavior, with positive effects on both sub-factors of food purchasing behavior: budgets (β-Value = 0.645; t-Value = 10.682) and supermarket offers (β-Value = 0.487; t-Value = 8.047). Therefore, Hypothesis 1 was accepted. Hypothesis 2 assumed that perceived food insecurity positively influences food preparation practices, and the results showed that perceived food insecurity had positive effects on meal planning (β-Value = 0.559; t-Value = 9.138) and resourcefulness (β-Value = 0.452; t-Value = 6.270). Hypothesis 3 established that food insecurity negatively influences satisfaction in life, which was also accepted (β-Value = −0.285; t-Value = −4.525). Lastly, Hypothesis 4 predicted that the influence of perceived food insecurity on food purchasing behavior, food preparation practices, and life satisfaction would vary depending on the generation. The results of an analysis that verified the moderating role of generations in this relationship are presented in Table 5. Considering the difference in df (∆ 2) between the unconstrained and constrained models, a difference of greater than 5.99 in chi-square values can be viewed as the presence of a significant moderating effect. According to the analysis results, the influence of perceived food insecurity on practical behavior in terms of resourcefulness differed depending on age, with the greatest influence on Generation M compared to the other generations. This suggests that a greater perceived food insecurity in Generation M would lead them to reduce the amount of food wasted and eat more meat-free meals to save money. In addition, the negative influence of perceived food insecurity on satisfaction in life was stronger in the older generation than in the younger generation; thus, Hypothesis 4 was partially accepted.

## 5. Discussion

This study examined the organic relationship between perceived food insecurity, food purchasing behavior, food preparation practices, and life satisfaction and confirmed the moderating role of generations in this relationship. According to the research results, perceived food insecurity positively influenced food purchasing behaviors and food preparation practices, which is also consistent with the findings of studies by Soldavini et al. [46], Miller et al. [49], and Ditlevsen et al. [52]—food insecurity worsens the quantity and quality of foods and increases the pattern of purchasing processed foods. In addition, this study verified the presence of a significant negative relationship between perceived food insecurity and satisfaction in life, which is aligned with the results of previous studies showing that food insecurity lowers life satisfaction [59,60,61]. Moreover, after reviewing the moderating effect of generations in the relationship between perceived food insecurity and food-related behavior and life satisfaction, a significant moderating role was found in some paths. Notably, a greater awareness of food insecurity in Generation M increased their amount of food wasted while increasing their behavior of eating meat-free meals to save money, and the negative impact of perceived food insecurity on satisfaction in life was more strongly exhibited in the older generation than in the younger generation.

Because food dominates human life, food insecurity is a critical global issue that should be promptly solved. At this point, the proposed result that food purchasing behaviors and food preparation practices and life satisfaction vary depending on the level of perceived food insecurity, along with the differential influence of generations on this relationship, provides multiple important implications for the government and communities. The theoretical implications derived from this study are as follows. First, most of the studies conducted thus far using the keyword food insecurity were confined to surveys of current status. At a time when research related to food insecurity and awareness of food crises is required in response to millions of people around the world suffering from hunger since the COVID-19 pandemic, the results of this study will have a significant timely value. Presenting the study results that food purchasing and preparation behaviors and life satisfaction can vary depending on the level of perceived food insecurity also provides a meaningful opportunity to raise our awareness of the risk of food crises and encourage additional research. Second, very few studies have examined the relationship between food crises or food insecurity as independent variables, with psychological and behavioral outcome variables. In particular, it is rare to find studies that identify the degree of perceived insecurity among the general public and its ripple effect. Consequently, the present study will not only shed light on the risks of food insecurity and explore the legitimacy of its negative impact on quality of life but also provide academic contributions to future research. In addition, this study tried to differentiate itself from existing studies focusing only on simple relations by introducing a unique moderating variable, generation, and pointed to the need to widen the current study. Furthermore, this study laid the foundation for further food insecurity studies to fully grasp the inherent suppositions about the negative effects of food insecurity.

The results of this study present important practical implications for food insecurity. The result that greater perceived food insecurity leads to corresponding more negative food purchasing and preparing behaviors and lowers life satisfaction will encourage the exploration of measures or policies for stabilizing food insecurity. The study results consequently highlight the urgent need for policy and government interventions to address the underlying economic factors that can contribute to resolving food insecurity. In particular, this study helps present policy measures that can raise awareness and seek solutions from a long-term perspective. Specifically, it presents the need to pay greater attention to food insecurity issues at the government level and accelerate related future research. Specifically, it will be necessary to establish a structured educational environment and support services for systematic food security education. It will also be necessary to find ways to maintain sustainable and healthy eating habits, even with limited resources, by educating the public on practical food use and preparation-related content. In addition, rather than educating people with high levels of perceived food insecurity about why they should eat healthy food, proactive efforts should first be made to remove barriers to healthy eating habits. As a preventive measure against the worsening of food insecurity, more solutions should be prepared from a macroscopic perspective in addition to implications from the microscopic perspective mentioned so far. For example, it is recommended to develop a realistic plan to purchase healthy food ingredients at reasonable prices through farmers’ markets or local food outlets by cooperating with local organizations and seeking institutional improvements for practically available systems, such as food vouchers, to help people purchase healthy food. This study also proved the moderating role of generations in the relationship between perceived food insecurity and food-related behaviors and life satisfaction, thereby verifying that the negative influence of food insecurity may vary depending on the generation. Specifically, the youngest generation showed the most prominent pattern of greater perceived food insecurity, leading to more thorough food preparation behaviors, such as reducing the amount of food wasted and money-saving behaviors. In terms of life satisfaction, the oldest generation exhibited the largest decline in life satisfaction caused by food insecurity. These results will help find differentiated ways to respond to food insecurity by generation, and to this end, it will be necessary to provide realistic implications for each age group. Specifically, since the younger generation is familiar with digital technology, information and policies related to food insecurity need to be actively promoted through digital media such as SNS, apps, and web content. It is also considered important to expand participatory policies through programs in which the younger generation can voluntarily participate, such as urban agriculture and food challenge movements.

When considering our results, several limitations of our study should be kept in mind. First, the study samples were relatively unevenly distributed due to their demographic characteristics, making it difficult to generalize the study results. There are limitations in terms of sample representativeness. Second, because the study was cross-sectional in nature and included only one-time measurements, longitudinal studies will be necessary in future studies. Third, although this study did not examine the antecedent factors that cause food insecurity, future research should be expanded to include antecedent variables, such as environmental factors. Fourth, this study presented food preparation and practice behaviors as food-related behaviors, but future research will need to consider food shopping-related behaviors. Fifth, this study divided the generations into Baby Boomers, Generation X, and Generation M, and it will be necessary to consider additional general characteristic variables (ex. occupation type, family size, and rural–urban residence) that influence food insecurity and food-related behaviors. These variables are not just background information, but key factors that cause or alleviate food insecurity. By analyzing them in an integrated manner, more sophisticated and effective food policies and health strategies can be developed. It would also be meaningful to demonstrate a comparative analysis with countries other than Korea. Comparative studies can help identify the strategies and policies of countries that are effectively alleviating food insecurity and adapt them to your own circumstances.

Sixth, since food insecurity is not simply a lack of food but a state in which it is difficult to achieve a healthy diet, it would also be meaningful to conduct research addressing the relationship with public health issues such as obesity and malnutrition. In conclusion, more practical research results will be possible if these limitations are improved in the future.

## 6. Conclusions

This study investigated the organic relationship among perceived food insecurity, food purchasing behavior, food preparation practices, and life satisfaction, and verified the moderating role of generation. The results of the study showed that perceived food insecurity had a positive effect on food purchasing behavior and food preparation practices, and that perceived food insecurity had a negative effect on life satisfaction. The results confirmed that there was a moderating role of generation in the effect of food insecurity on food preparation practices and life satisfaction. The results of this study are expected to provide important implications for the study of food-related behavior related to perceived food insecurity. In summary, this study shows that food insecurity is not a simple problem of deficiency, but a complex factor that affects individuals’ food purchasing and preparation behaviors and overall quality of life. In particular, the fact that these effects differ by generation suggests the need for a generation-specific approach when designing food insecurity response policies and education programs.

## Figures and Tables

**Figure 1 ijerph-22-00766-f001:**
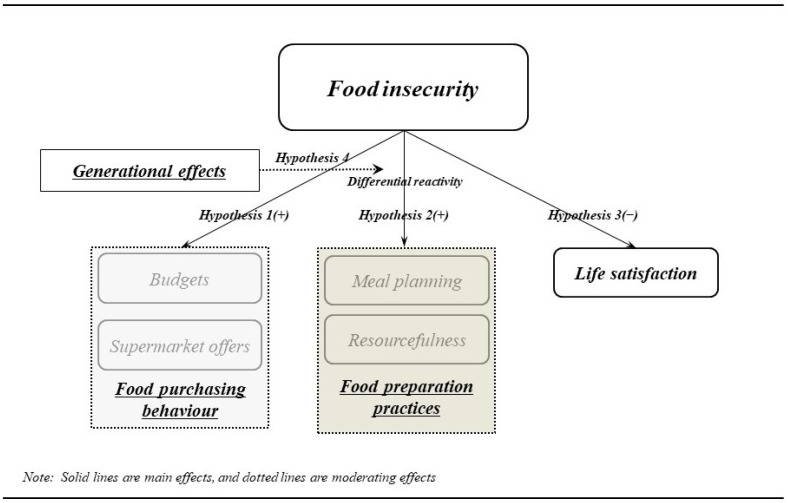
Research model.

**Table 1 ijerph-22-00766-t001:** Profile of the sample (N = 300).

Characteristic	N	Percentage
Gender		
Male	156	52.0
Female	144	48.0
Generations		
M generation	116	38.7
X generation	106	35.3
Boomer generation	78	26.0
Education level		
High school	21	7.0
College	81	27.0
University	198	66.0
Marital status		
Married	165	55.0
Unmarried	128	42.6
Others	7	2.4

**Table 2 ijerph-22-00766-t002:** Confirmatory factor analysis and reliability analysis.

Construct (Cronbach’s Alpha)	Standardized Estimate	t-Value	CCR ^a^	AVE ^b^
Food insecurity (0.940)			0.864	0.689
FI1	0.894	Fixed		
FI2	0.900	23.739 ***		
FI3	0.849	20.839 ***		
FI4	0.830	19.854 ***		
FI5	0.761	16.924 ***		
FI6	0.811	18.970 ***		
FI7	0.755	16.688 ***		
Food purchasing behavior: Budgets (0.966)			0.925	0.828
BU1	0.885	Fixed		
BU2	0.938	26.395 ***		
BU3	0.949	27.190 ***		
BU4	0.912	24.523 ***		
BU5	0.917	24.873 ***		
BU6	0.858	21.298 ***		
Food purchasing behavior: Supermarket offers (0.934)			0.849	0.782
SO1	0.924	Fixed		
SO2	0.870	23.122 ***		
SO3	0.889	24.318 ***		
SO4	0.854	22.124 ***		
Food preparation practices: Meal planning (0.844)			0.708	0.655
EA1	0.949	Fixed		
EA2	0.786	15.937 ***		
EA3	0.671	12.949 ***		
Food preparation practices: Resourcefulness (0.801)			0.712	0.597
RE1	0.620	Fixed		
RE2	0.880	10.687 ***		
RE3	0.796	10.498 ***		
Life satisfaction (0.839)			0.746	0.662
LS1	0.787	Fixed		
LS2	0.665	12.170 ***		
LS3	0.963	14.515 ***		

Note: ^a^ CCR = composite construct reliability; ^b^ AVE = average variance extracted; Standardized estimate = β-value; χ^2^ = 633.999 (df = 284) *p* < 0.001; χ^2^/df = 2.232; Goodness-of-Fit Index (GFI) = 0.858; Tucker Lewis Index (TLI) = 0.941; Comparative Fit Index (CFI) = 0.948; Incremental Fit Index (IFI) = 0.949; Root Square Error of Approximation (RMSEA) = 0.064; *** *p* < 0.001.

**Table 3 ijerph-22-00766-t003:** Means, standard deviations, and correlation analyses.

Construct	1	2	3	4	5	Mean ± SD ^a^
Food insecurity	1					2.82 ± 1.34
Budgets	0.592 **	1				3.02 ± 1.41
Supermarket offers	0.423 **	0.471 **	1			4.17 ± 1.46
Meal planning	0.471 **	0.440 **	0.369 **	1		2.90 ± 1.30
Resourcefulness	0.403 **	0.413 **	0.329 **	0.325 **	1	3.97 ± 1.23
Life satisfaction	−0.208 **	−0.235 **	−0.249 **	−0.126 *	−0.134 *	4.31 ± 1.19

Note: ^a^ SD = Standard Deviation; All variables were measured on a 7-point Likert scale from 1 (strongly disagree) to 7 (strongly agree), ** *p* < 0.01; * *p* < 0.05.

**Table 4 ijerph-22-00766-t004:** Structural parameter estimates.

Hypothesized Path (Stated as Alternative Hypothesis)	Standardized Path Coefficients	t-Value	Results
H1a: Food insecurity → Budgets	0.645	10.682 ***	Accepted
H1b: Food insecurity → Supermarket offers	0.487	8.047 ***	Accepted
H2a: Food insecurity → Meal planning	0.559	9.138 ***	Accepted
H2b: Food insecurity → Resourcefulness	0.452	6.270 ***	Accepted
H3: Food insecurity → Life satisfaction	−0.285	−4.525 ***	Accepted
Goodness-of-fit statistics	χ^2^(294) = 715.164 (*p* < 0.001)		
	χ^2^/df = 2.433		
	GFI = 0.838		
	IFI = 0.938		
	CFI = 0.938		
	RMSEA = 0.069		

Note: *** *p* < 0.001; GFI = Goodness-of-Fit Index; NFI = Normed Fit Index; CFI = Comparative Fit Index; RMSEA = Root Mean Square Error of Approximation.

**Table 5 ijerph-22-00766-t005:** Moderating role of Generations M, X, and Boomer.

	M Generation (N = 116)		X Generation (N = 106)		Boomer Generation (N = 78)		Unconstrained Model Chi-Square (df = 882)	Constrained Model Chi-Square (df = 884)	**∆χ** ** ^2^ ** **(df = 2)**
	Standardized Coefficients	t-Value	Standardized Coefficients	t-Value	Standardized Coefficients	t-Value			
H4a: Food insecurity → Budgets	0.652	6.388 ***	0.692	6.223 ***	0.563	5.324 ***	1620.291	1625.244	4.953 ^ns^
H4b: Food insecurity → Supermarket offers	0.427	4.310 ***	0.429	4.037 ***	0.656	6.395 ***		1620.555	0.264 ^ns^
H4c: Food insecurity → Meal planning	0.640	6.068 ***	0.463	4.473 ***	0.524	4.692 ***		1624.188	3.897 ^ns^
H4d: Food insecurity → Resourcefulness	0.579	4.877 ***	0.221	1.983 *	0.492	3.315 ***		1626.756	6.465 *
H4e: Food insecurity → Life satisfaction	−0.158	−1.580 ns	−0.355	−3.382 ***	−0.392	−3.287 **		1626.315	6.024 *

Note: χ^2^/df = 1.837; NFI = 0.802; TLI = 0.887; CFI = 0.898; RMSEA = 0.053; * *p* < 0.05, ** *p* < 0.01, *** *p* < 0.001, ^ns^ Not significant.

## Data Availability

The data presented in this study are available on request from the corresponding author.

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
