# Peer review of "Is the Impact of Food Insecurity on Food-Related Behavior Moderated by Generation in South Korea?"

_ijerph, 2025, doi:10.3390/ijerph22050766_

Round 1

Reviewer 1 Report

Comments and Suggestions for Authors

In the "Introduction" and "Literature Review" sections, update the references to include more recent studies on how post-pandemic economic recovery and new trade policies affect food insecurity, not just the immediate impacts during the pandemic. Provide more in-depth analysis of the South Korean context, such as unique trade policies, cultural attitudes toward food, or recent changes in the domestic food production system.In the "Methodology" section, diversify the sampling method to avoid bias from using only consumers of a discount platform. Consider adding samples from different income levels, regions, and shopping habits by collaborating with community centers or using stratified random sampling. For the language, simplify complex sentences and ensure consistent terminology. In the "Discussion and Conclusions" section, expand on limitations by explaining how uneven sample distribution might affect results. Be more specific about future research directions, such as listing potential variables like occupation type, family size, or rural-urban residence, and suggesting which countries and aspects to compare, like the impact of different welfare systems on food-related behaviors across generations.

Comments on the Quality of English Language

The English in this paper is functional, it has areas that need improvement to better convey the research and enhance readability. There are grammar issues, like in the abstract where “Notably, a greater awareness of food insecurity in Generation M increased resourcefulness preparation practices” is unclear due to the term “resourcefulness preparation practices” and could be rephrased. Sentence flow problems exist, such as in the introduction with “However, many social and environmental events threatening secure food systems continue to occur globally today [3], which makes the public highly vulnerable to food insecurity problems [4]” which can be simplified. Vocabulary usage can be more precise, like replacing “food cooking skills” with “culinary skills” in the literature review. Also, there's a lack of consistency in hyphen use for compound words like “food - related” and “food related”. Addressing these language issues will make the paper more accessible to a broader international audience and more effectively communicate its significant research findings.

Author Response

Reviewer 1

We would like to thank the international panel of reviewer for their thoughtful consideration of our manuscripts. We appreciate the time and effort you put into reviewing our manuscript. Reviewer provided valuable guidelines for improving the paper.

____________________

  1. In the "Introduction" and "Literature Review" sections, update the references to include more recent studies on how post-pandemic economic recovery and new trade policies affect food insecurity, not just the immediate impacts during the pandemic. → We appreciate Reviewer’s precise comment. As mentioned, we revised it.

  1. Provide more in-depth analysis of the South Korean context, such as unique trade policies, cultural attitudes toward food, or recent changes in the domestic food production system. → We highly value Reviewer’s comment on this issue. Updated on the current situation and conditions in South Korea.

  1. In the "Methodology" section, diversify the sampling method to avoid bias from using only consumers of a discount platform. Consider adding samples from different income levels, regions, and shopping habits by collaborating with community centers or using stratified random sampling. → We highly value Reviewer’s comment on this issue. As suggested, we revised this part.The sampling method was discussed in more detail.

  1. For the language, simplify complex sentences and ensure consistent terminology. → Thank you for your valuable advice. Overall, it has been revised to be as unified as possible.

  1. In the "Discussion and Conclusions" section, expand on limitations by explaining how uneven sample distribution might affect results. Be more specific about future research directions, such as listing potential variables like occupation type, family size, or rural-urban residence, and suggesting which countries and aspects to compare, like the impact of different welfare systems on food-related behaviors across generations. → We highly value Reviewer’s comment on this issue. These parts have been supplemented overall to the limit.

  1. The English in this paper is functional, it has areas that need improvement to better convey the research and enhance readability. There are grammar issues, like in the abstract where “Notably, a greater awareness of food insecurity in Generation M increased resourcefulness preparation practices” is unclear due to the term “resourcefulness preparation practices” and could be rephrased. Sentence flow problems exist, such as in the introduction with “However, many social and environmental events threatening secure food systems continue to occur globally today [3], which makes the public highly vulnerable to food insecurity problems [4]” which can be simplified. Vocabulary usage can be more precise, like replacing “food cooking skills” with “culinary skills” in the literature review. Also, there's a lack of consistency in hyphen use for compound words like “food - related” and “food related”. Addressing these language issues will make the paper more accessible to a broader international audience and more effectively communicate its significant research findings.  → We appreciate Reviewer’s precise comment. We have revised the parts you pointed out.  We have checked the entire text to ensure consistency and unity of terminology.

References added for the revised manuscript

  1. Paslakis, G.; Dimitropoulos, G.; Katzman, D. K. A call to action to address COVID-19-induced global food insecurity to prevent hunger, malnutrition, and eating pathology. Nutrition Reviews 2021, 79, 114–116.
  2. Kakaei, H.; Nourmoradi, H.; Bakhtiyari, S.; Jalilian,; Mirzaei, A. Effect of COVID-19 on food security, hunger, and food crisis. COVID-19 and the Sustainable Development Goals 2022, 29, 3–29. 
  3. Wolfson, J. ; Leung, C. W. Food Insecurity in the COVID-19 Era: A National Wake-up Call to Strengthen SNAP Policy. Annals of Internal Medicine 2024, 177, 255-256.
  4. Kumar, N.; Quisumbing, A. Gendered impacts of the 2007–2008 food price crisis: evidence using panel data from rural Ethiopia. Food Policy 2013, 38, 11-22.
  5. Mane, E.; Giaquinto, A. ; Cafiero, C.; Viviani, S.; Anriquez, G. Closing the gender gap in global food insecurity: Socioeconomic determinants and economic gains in the aftermath of COVID-19. Global Food Security 2025, 45, 100850.
  6. Choe, H.; Pak, T. Food insecurity and unmet healthcare needs in South Korea. International Journal for Equity in Health 2023,22, 148.
  7. Baek, S. ; Yoon, J. H. The mediating role of food insecurity in the relationship between income poverty and depressive symptoms and suicidal ideation: A nationwide study of Korean adults. Social Science & Medicine 2025, 373, 117972.

Reviewer 2 Report

Comments and Suggestions for Authors

Dear authors, 

The manuscript analyses the relation between food insecurity and food-related behaviors and life satisfaction using data from South Korea. It is an interesting topic that needs further research. 

My major concerns:

  1. The abstract uses the word "causal". The methodology assumes rather the empirical test for causality. However, someone can argue the inverse relation, such as, cooking practice may also causes food insecurity.  Therefore, I would suggest: (i) add a discussion based on previous research / references to argue that food insecurity leads to food-related behaviors (and not the other way around). (ii) avoid using "causal" arguments, rather use "association" arguments.
  2. Some on the conclusions are not directed linked with the results (in the abstract and discussion/conclusions sections). Therefore, please keep your discussion related with the results (no policy recommendations without empirical support) 
  3. The manuscript uses the work by Myers (2020) to assess food insecurity. However, this work does not develop an instrument to measure food insecurity. Why not use the FIES by FAO?

Minors comments:

  1. There are some minors comments (added on the side in the MS Word version) that need to be addressed.
  2. The manuscript uses a sample of 300 participants, how different is from the overall Korean population? I understand that is not a representative sample, however, it would be informative to know how different is. 

Good luck!!

Comments on the Quality of English Language

English is my second language. I read it ok. No comments.

Author Response

Reviewer 2

We would like to thank the international panel of reviewer for their thoughtful consideration of our manuscripts. We appreciate the time and effort you put into reviewing our manuscript. Reviewer provided valuable guidelines for improving the paper.

  1. The abstract uses the word "causal". The methodology assumes rather the empirical test for causality. However, someone can argue the inverse relation, such as, cooking practice may also causes food insecurity.  Therefore, I would suggest: (i) add a discussion based on previous research / references to argue that food insecurity leads to food-related behaviors (and not the other way around). (ii) avoid using "causal" arguments, rather use "association" arguments. Thank you for your comment. We revised it.  The entire text was revised in the context of correlation (association) analysis.

  1. Some on the conclusions are not directed linked with the results (in the abstract and discussion/conclusions sections). Therefore, please keep your discussion related with the results (no policy recommendations without empirical support)  We highly value Reviewer’s comment on this issue.  The discussion and conclusion were written separately, and the parts that made policy recommendations without empirical evidence were revised.

  1. The manuscript uses the work by Myers (2020) to assess food insecurity. However, this work does not develop an instrument to measure food insecurity. Why not use the FIES by FAO?  We appreciate Reviewer’s meticulous comment.  Thanks for the good suggestion. Many studies have used the Meyers (2020) scale. We will consider the FAO scale in future studies. 

  1. The manuscript uses a sample of 300 participants, how different is from the overall Korean population? I understand that is not a representative sample, however, it would be informative to know how different is. We appreciate Reviewer’s precise comment. The survey was conducted by commissioning a credible survey company, and additional improvements were made to address these limitations.

References added for the revised manuscript

  1. Paslakis, G.; Dimitropoulos, G.; Katzman, D. K. A call to action to address COVID-19-induced global food insecurity to prevent hunger, malnutrition, and eating pathology. Nutrition Reviews 2021, 79, 114–116.
  2. Kakaei, H.; Nourmoradi, H.; Bakhtiyari, S.; Jalilian,; Mirzaei, A. Effect of COVID-19 on food security, hunger, and food crisis. COVID-19 and the Sustainable Development Goals 2022, 29, 3–29. 
  3. Wolfson, J. ; Leung, C. W. Food Insecurity in the COVID-19 Era: A National Wake-up Call to Strengthen SNAP Policy. Annals of Internal Medicine 2024, 177, 255-256.
  4. Kumar, N.; Quisumbing, A. Gendered impacts of the 2007–2008 food price crisis: evidence using panel data from rural Ethiopia. Food Policy 2013, 38, 11-22.
  5. Mane, E.; Giaquinto, A. ; Cafiero, C.; Viviani, S.; Anriquez, G. Closing the gender gap in global food insecurity: Socioeconomic determinants and economic gains in the aftermath of COVID-19. Global Food Security 2025, 45, 100850.
  6. Choe, H.; Pak, T. Food insecurity and unmet healthcare needs in South Korea. International Journal for Equity in Health 2023,22, 148.
  7. Baek, S. ; Yoon, J. H. The mediating role of food insecurity in the relationship between income poverty and depressive symptoms and suicidal ideation: A nationwide study of Korean adults. Social Science & Medicine 2025, 373, 117972.

Reviewer 3 Report

Comments and Suggestions for Authors

I have to point out that this study's conclusions should be formulated in a separate chapter.

In this chapter, conclusions should be formulated differently. Firstly, the authors have to make clear the meaning of the conclusions, and secondly, write down the limitations and the conditions that have arisen. 

Generously is an interesting scientific work that implements CFA and SEM on data collected from questionnaires. These questionnaires should be described in more details in the chapter of Methodology

Author Response

Reviewer 3

We would like to thank the international panel of reviewer for their thoughtful consideration of our manuscripts. We appreciate the time and effort you put into reviewing our manuscript. Reviewer provided valuable guidelines for improving the paper.

  1. I have to point out that this study's conclusions should be formulated in a separate chapter. Thank you for your reasonable advice. The conclusion is described in a separate section.

  1. In this chapter, conclusions should be formulated differently. Firstly, the authors have to make clear the meaning of the conclusions, and secondly, write down the limitations and the conditions that have arisen.  Thank you for kind comment. The Conclusions and Discussion sections have been revised throughout.
  2. Generously is an interesting scientific work that implements CFA and SEM on data collected from questionnaires. These questionnaires should be described in more details in the chapter of Methodology. Thank you for your reasonable advice. The contents of the questionnaire were mentioned as sample questions.

References added for the revised manuscript

  1. Paslakis, G.; Dimitropoulos, G.; Katzman, D. K. A call to action to address COVID-19-induced global food insecurity to prevent hunger, malnutrition, and eating pathology. Nutrition Reviews 2021, 79, 114–116.
  2. Kakaei, H.; Nourmoradi, H.; Bakhtiyari, S.; Jalilian,; Mirzaei, A. Effect of COVID-19 on food security, hunger, and food crisis. COVID-19 and the Sustainable Development Goals 2022, 29, 3–29. 
  3. Wolfson, J. ; Leung, C. W. Food Insecurity in the COVID-19 Era: A National Wake-up Call to Strengthen SNAP Policy. Annals of Internal Medicine 2024, 177, 255-256.
  4. Kumar, N.; Quisumbing, A. Gendered impacts of the 2007–2008 food price crisis: evidence using panel data from rural Ethiopia. Food Policy 2013, 38, 11-22.
  5. Mane, E.; Giaquinto, A. ; Cafiero, C.; Viviani, S.; Anriquez, G. Closing the gender gap in global food insecurity: Socioeconomic determinants and economic gains in the aftermath of COVID-19. Global Food Security 2025, 45, 100850.
  6. Choe, H.; Pak, T. Food insecurity and unmet healthcare needs in South Korea. International Journal for Equity in Health 2023,22, 148.
  7. Baek, S. ; Yoon, J. H. The mediating role of food insecurity in the relationship between income poverty and depressive symptoms and suicidal ideation: A nationwide study of Korean adults. Social Science & Medicine 2025, 373, 117972.

Round 2

Reviewer 1 Report

Comments and Suggestions for Authors

The revised manuscript shows improvement, but there's room for further refinement. Deepen the discussion by incorporating relevant sociological theories to better explain generational differences in responses to food insecurity. Expand the practical implications by offering more specific and actionable recommendations for various stakeholders, like detailed policy measures for policymakers and community - based initiatives for community organizations. In the limitations section, clarify the potential impact of remaining limitations on the findings and explore mitigation strategies. Finally, strengthen the conclusion by concisely summarizing key contributions and highlighting long - term implications for food security research and social policy.

Comments on the Quality of English Language

The English in the revised manuscript has improved, but it can be further enhanced. Some sentences are still complex and could be simplified for better flow, especially in sections with intricate variable - relationship discussions. The use of jargon should be more accessible; briefly explain or provide synonyms for less - common terms. Additionally, minor grammar errors like subject - verb agreement in complex clauses remain. A proofread by a native English speaker or professional editor would help address these issues, making the research more engaging and understandable globally.

Author Response

We would like to thank the international panel of reviewer for their thoughtful consideration of our manuscripts. We appreciate the time and effort you put into reviewing our manuscript. Reviewer provided valuable guidelines for improving the paper.
